# Metformin Suppresses Cancer Stem Cells through AMPK Activation and Inhibition of Protein Prenylation of the Mevalonate Pathway in Colorectal Cancer

**DOI:** 10.3390/cancers12092554

**Published:** 2020-09-08

**Authors:** Yoojeong Seo, Janghyun Kim, Soo Jung Park, Jae Jun Park, Jae Hee Cheon, Won Ho Kim, Tae Il Kim

**Affiliations:** 1Institute of Gastroenterology, Yonsei University College of Medicine, Seoul 03722, Korea; yjseo90@yuhs.ac (Y.S.); kimsbora2013@yuhs.ac (J.K.); sjpark@yuhs.ac (S.J.P.); jaejpark@yuhs.ac (J.J.P.); geniushee@yuhs.ac (J.H.C.); kimwonho@yuhs.ac (W.H.K.); 2Brain Korea 21 PLUS Project for Medical Science, Yonsei University College of Medicine, Seoul 03722, Korea; 3Department of Internal Medicine, Yonsei University College of Medicine, Seoul 03722, Korea; 4Cancer Prevention Center, Yonsei University College of Medicine, Seoul 03722, Korea

**Keywords:** cancer stem cells, metformin, mevalonate pathway, protein prenylation, colorectal cancer

## Abstract

**Simple Summary:**

Tumor suppressing effect of metformin has been reported, and one of mechanism of this effect is suppression of cancer stem cells (CSCs). However, detailed mechanism of metformin-induced CSC-inhibitory effect has not been known. We demonstrated that the CSC-suppressive effect of metformin was associated with AMPK activation/mTOR inhibition and repression of protein prenylation through suppression of mevalonate pathway in colorectal cancer. Further studies would be needed to investigate cross-reactions with other mechanisms of the antitumor effect of metformin, and clinical impact of metformin should be considered as chemopreventive or adjunctive treatment for colorectal tumor.

**Abstract:**

Metformin is a well-known AMPK (AMP-activated protein kinase) activator that suppresses cancer stem cells (CSCs) in some cancers. However, the mechanisms of the CSC-suppressing effects of metformin are not yet well understood. In this study, we investigated the CSC-suppressive effect of metformin via the mevalonate (MVA) pathway in colorectal cancer (CRC). Two colorectal cancer cell lines, HT29 and DLD-1 cells, were treated with metformin, mevalonate, or a combination of the two. We measured CSC populations by flow cytometric analysis (CD44+/CD133+) and by tumor spheroid growth. The expression of p-AMPK, mTORC1 (pS6), and key enzymes (HMGCR, FDPS, GGPS1, and SQLE) of the MVA pathway was also analyzed. We investigated the effects of metformin and/or mevalonate in xenograft mice using HT29 cells; immunohistochemical staining for CSC markers and key enzymes of the MVA pathway in tumor xenografts was performed. In both HT29 and DLD-1 cells, the CSC population was significantly decreased following treatment with metformin, AMPK activator (AICAR), HMG-CoA reductase inhibitor (simvastatin), or mTOR inhibitor (rapamycin), and was increased by mevalonate. The CSC-suppressing effect of these drugs was attenuated by mevalonate. The results of tumor spheroid growth matched those of the CSC population experiments. Metformin treatment increased p-AMPK and decreased mTOR (pS6) expression; these effects were reversed by addition of mevalonate. The expression of key MVA pathway enzymes was significantly increased in tumor spheroid culture, and by addition of mevalonate, and decreased upon treatment with metformin, AICAR, or rapamycin. In xenograft experiments, tumor growth and CSC populations were significantly reduced by metformin, and this inhibitory effect of metformin was abrogated by combined treatment with mevalonate. Furthermore, in the MVA pathway, CSC populations were reduced by inhibition of protein prenylation with a farnesyl transferase inhibitor (FTI-277) or a geranylgeranyl transferase inhibitor (GGTI-298), but not by inhibition of cholesterol synthesis with a squalene synthase inhibitor (YM-53601). In conclusion, the CSC-suppressive effect of metformin was associated with AMPK activation and repression of protein prenylation through MVA pathway suppression in colorectal cancer.

## 1. Introduction

Colorectal cancer (CRC) is the third most common cause of cancer-related deaths worldwide [1]. Approximately 25% of CRC patients are already stage IV at diagnosis, and a significant proportion of advanced CRC patients develop resistance to treatment and experience recurrence and metastasis [2]. Therefore, to develop more effective drugs for advanced CRC, we need to target the critical cells and signaling pathways involved in tumor progression, recurrence, and metastasis.

Cancer stem cells (CSCs) are a main target for decreasing cancer risk, recurrence, and metastasis and for overcoming resistance to chemotherapy. CSCs are maintained as a subpopulation within the neoplasm that perpetuates clonal expansion. This subpopulation generally has resistance to conventional chemotherapy, leading to relapse and metastasis. Therefore, many researchers have worked to identify critical pathways for maintenance of CSCs and to devise CSC-targeted or CSC-differentiating therapy [3,4].

Metformin (*N*′, *N*′-dimethylbiguanide) is a well-known oral antidiabetic drug that activates AMP-dependent protein kinase (AMPK). In clinical studies, metformin decreases colon polyp development, as well as the risk, recurrence, and mortality of CRC [5].

These beneficial effects of metformin are related to AMPK activation, mammalian target of rapamycin (mTOR) inhibition, cell cycle regulation, and CSC suppression [6,7,8,9].

Recent studies reported additional potent effects of metformin on lipid and cholesterol biosynthesis in colorectal cancer cells. Metformin inhibits transcriptional activation of β-hydroxy-β-methylglutaryl-CoA (HMG-CoA) reductase, as well as key mediators in the cholesterol biosynthesis pathway [10,11,12]. The cholesterol biosynthesis/mevalonate pathway, also known as the HMG-CoA reductase pathway, is critical for cancer cell survival [13]. In clinical studies, elevated cholesterol is strongly associated with high cancer risk, worse outcomes, and resistance to chemotherapy [14,15], and statin, a cholesterol-lowering agent and inhibitor of HMG-CoA reductase, decreases cancer risk and has antitumor effects [16].

The mevalonate (MVA) pathway leads to the synthesis of sterols and protein prenylation that have important roles in tumor growth. Multiple enzymes of this pathway are essential for proliferation and survival of various types of cancer cells, and are upregulated in many cancers [15,17]. In addition, key final enzymes of the MVA pathway, including farnesyl diphosphate synthase (FDPS), geranylgeranyl pyrophosphate synthase 1 (GGPS1), and squalene epoxidase (SQLE), have been linked to activation of farnesylated and geranylgeranylated proteins, such as Ras and Ral/Rho, and cholesterol synthesis, respectively.

However, there are no data that elucidate the direct relationship between metformin-induced suppression of CSCs and its detailed mechanism in the MVA pathway. Here, we showed that CSC suppression by metformin is associated with inhibition of protein prenylation, rather than cholesterol synthesis, through the MVA pathway.

## 2. Results

### 2.1. Metformin, an AMPK Activator and mTOR Inhibitor, Suppresses Cancer Stem Cells

Based on our previously reported data [8], we selected HT29 and DLD-1 cells from the available metformin-sensitive CRC cell lines. We confirmed decreased mRNA expression of the CSC markers Lgr5, CD44, and CD133 (Figure 1A).

In addition, in another public transcript data set (GSE76342) for the colon cancer cell line LoVo with or without metformin treatment, we found that metformin inhibited expression of the CSC markers Lgr5, ASCL2, EPHB3, OLFM4, BMI1, Lrig1, TERT, CD44, and CD133 (Figure 1B). Because one of the well-known pathways of metformin action is the AMPK-dependent mTOR pathway, we examined the expression of activated AMPK and mTOR signaling molecules after treatment with metformin. Metformin treatment increased p-AMPK and decreased p-S6 expression (Figure 1C). Using flow cytometric analysis, we confirmed that the CSC population was significantly decreased by metformin, AICAR (AMPK activator), simvastatin (HMG-CoA reductase inhibitor), and rapamycin (mTOR inhibitor) (Figure 1D). In addition, in tumor sphere culture experiments, tumor sphere formation was significantly decreased by treatment with these drugs (Figure 1E,F).

### 2.2. Metformin Suppresses Key Enzymes of the Mevalonate Pathway

Some reports have shown potent effects of metformin on lipid and cholesterol biosynthesis in cancer cells [11,12]. Therefore, we postulated that metformin may inhibit HMG-CoA reductase (HMGCR), as well as several important enzymes of the MVA pathway, in CRC. In the same data set shown in Figure 1B, we found that metformin reduced key enzymes of the MVA pathway, including HMGCR, mevalonate kinase (MVK), phospho-mevalonate kinase (PMVK), mevalonate decarboxylase (MVD), FDPS, GGPS, and SQLE (Figure 2A).

To demonstrate increased activity of the MVA pathway in CSCs of CRC, we cultured tumor spheres and found significantly elevated protein and mRNA levels of key enzymes of the MVA pathway, including HMGCR, FDPS, GGPS1, and SQLE, in tumor spheroids compared to 2D adherent cultured cells (Figure 2B,C). These findings suggest that the MVA pathway may have an important role in CRC tumorigenesis by promoting CSCs of CRC. In addition, metformin significantly reduced expression of key enzymes of the MVA pathway that were upregulated in 3D tumor spheroid cultures of CRC cells (Figure 2B,C).

We asked whether other AMPK/mTOR regulators can inhibit enzymes of the MVA pathway. Thus, we confirmed significant decreases in mRNA levels of these key enzymes of the MVA pathway in tumor spheres after 7 days of treatment with metformin, AICAR, or rapamycin. We also tested simvastatin on tumor spheres of CRC cells, and found that reduction of these key enzymes of the MVA pathway by simvastatin was relatively weak and inconsistent compared to the effects of metformin (Figure 2D). Moreover, treatment with mevalonate promoted expression of these key enzymes of the MVA pathway (Figure 2D), resulting in an increased proportion of CSCs among CRC cells (Figure 2E). In addition, mevalonate treatment attenuated the suppressive effect of metformin on CSCs of CRC and the formation of tumor spheroids (Figure 2E,F). This reversal effect of mevalonate was also observed in treatment with AICAR, simvastatin, and rapamycin in both cell lines (Figure 2E).

Taken together, these results demonstrate that the MVA pathway was activated in CSCs and that metformin, AMPK activation, and mTOR inhibition suppressed CSCs through inhibiting the expression of key enzymes of the MVA pathway.

### 2.3. Metformin Reduced CSCs through Inhibition of Protein Prenylation, rather than Cholesterol Synthesis, in the MVA Pathway

We further investigated the specific final processes of the MVA pathway associated with metformin-induced suppression of CSCs by performing experiments using inhibitors of protein prenylation, such as farnesylation and geranylgeranylation, and of cholesterol synthesis. We used the farnesyl transferase inhibitor FTI-277, the geranylgeranyl transferase inhibitor GGTI-298, and the squalene synthase inhibitor YM-53601. To evaluate the CSC-specific effects and exclude the possibility of direct cellular toxicity of these inhibitors, we selected concentrations of inhibitors showing CSC-suppressive effects without significant cell death. For example, higher concentrations of YM-53601 (>10 μM) induced significant cell death without changes in CSC population (data not shown). FTI-277 and GGTI-298 induced suppression of the CSC population in a dose-dependent manner, similar to metformin treatment (Figure 3A). Moreover, combined treatment of FTI-277 and GGTI-298 decreased the CSC population even further (Figure 3A). However, YM-53601 did not show a significant effect on CSC populations (Figure 3A). Furthermore, in tumor sphere formation assays (Figure 3B,C), treatment with FTI-277, GGTI-298, or YM-53601 produced results similar to those observed in flow cytometric analyses of CSC populations. In addition, to confirm the decrease of protein prenylation by metformin, we performed Western blot analysis to identify the change of shifted prenylated protein bands of RAS and Ral A by farnesylation and geranylgeranylation, respectively. Although the shifted bands were weak in some conditions, we found the decrease of shifted prenylated protein bands of RAS and Ral A (Appendix A). These results demonstrated that metformin may suppress CSCs through inhibition of protein prenylation, via farnesylation and geranylgeranylation, rather than through cholesterol synthesis by SQLE.

### 2.4. Tumor-Suppressing Effect of Metformin in a Murine Xenograft Model Was Reversed by Additional Treatment with Mevalonate

We performed in vivo xenograft experiments to evaluate the effect of mevalonate on metformin-induced tumor suppression. At 14 days after implantation of HT29 cells, we treated mice with mevalonate, metformin, or a combination of mevalonate and metformin, by intraperitoneal injection for 21 days (Figure 4A). In the metformin-treated group, tumor growth was suppressed by 20% compared to the control group. Treatment with mevalonate alone showed a trend toward further tumor growth relative to the control group; the addition of mevalonate to metformin treatment induced a significant increase in tumor growth compared to metformin treatment alone, indicating that mevalonate could reverse the tumor-suppressing effect of metformin (Figure 4B–D). In immunohistochemical (IHC) staining for the CSC markers CD44 and CD133, the metformin treatment group showed significantly decreased IHC scores for CD44 and CD133; combination treatment of metformin and mevalonate induced significant increases in both CSC markers compared to metformin treatment alone (Figure 4E,F). Moreover, key enzymes of protein prenylation of the MVA pathway, FDPS and GGPS1, were suppressed by metformin, and this suppressive effect was reversed by addition of mevalonate (Figure 4G,H). From these results, we confirmed that mevalonate could reverse the CSC-suppressing effect of metformin in the in vivo xenograft model, and metformin-induced CSC suppression could be associated with suppression of key enzymes of the MVA pathway. In addition, we performed Ki67 staining to show antiproliferative effect of metformin, and found a significant decrease of Ki67 staining by treatment of metformin (Appendix A). Therefore, the tumor-suppressing effect of metformin would not depend on only suppression of CSCs. However, because the CSCs are important for tumor initiation, persistent progression, and metastasis, and the effect of metformin was more prominent on CSC population in our previous data [8], we focused on CSC suppression by metformin.

## 3. Discussion

Several previous studies revealed the inhibitory effect of metformin on CSCs [8,9], but the detailed mechanism has not been well defined. In this study, we identified that metformin serves as a negative regulator of the MVA pathway and exerts CSC-suppressive effects in CRC through the inhibition of protein prenylation processes of the MVA pathway. Regarding the relationship among tumor growth, the cholesterol pathway, and metformin, Sharma at al. showed increased cholesterol levels and elevated expression of certain cholesterol regulatory genes in malignant breast tumor tissues, such as HMGCR, LDLR, and SREBP1 [12]. They also demonstrated that metformin inhibited cholesterol levels in breast cancer MDA-MB-231 cells, with decreased expression of cholesterol regulatory genes. In addition, Wahdan-Alaswad et al. suggested that metformin suppressed triple-negative MDA-MB-231 cells by reducing fatty acid synthesis [10,11]. Furthermore, they showed that metformin targets the cholesterol biosynthesis pathway and stabilizes GM1 lipid rafts, reducing membrane EGFR (epidermal growth factor receptor) signaling and its activation in triple-negative breast cancer. Moreover, they demonstrated that the combination of metformin with the statin-mimetic methyl-β-cyclodextrin (MβCD) synergistically attenuates cholesterol biosynthesis and cell proliferation. In particular, they suggested that metformin might act as an HMGCR inhibitor. In our analysis using their public RNA-seq data for the metformin-treated cell line, we found that expression of most genes involved in the MVA pathway was reduced by metformin treatment (Figure 2A). Several other studies also demonstrated that metformin may inhibit HMGCR and SREBP1 in different cell types [18,19,20]. However, the detailed relationship between metformin-induced regulation of the MVA/cholesterol pathway and CSCs has not been defined.

Our study aimed to investigate how metformin affects the MVA pathway and whether this may be related to the inhibitory effect on CSCs. We showed that metformin inhibited CSCs in CRC cells, with concomitant decreases in MVA pathway enzymes, including HMGCR, FDPS, GGPS1, and SQLE. We observed that the inhibitory effect of metformin on CSCs and MVA pathway enzymes was reversed by addition of mevalonate. The elevated expression of these MVA pathway enzymes was related to increased CSCs of CRC, suggesting a close relationship between CSCs and activated MVA pathway.

Interestingly, we demonstrated that CSCs could be reduced through the inhibition of protein prenylation, but not by inhibition of cholesterol synthesis. The prenylation of Ras and Ral/Rho proteins via MVA pathway-dependent farnesylation and geranylgeranylation, respectively, is important for their function. The Ras, Ral, and Rho families are associated with many tumor characteristics, such as invasive growth, cell survival, and three-dimensional growth, [21,22], and play critical roles in tumor development, progression, and metastasis, including activation of CSCs [23,24,25]. Therefore, inhibition of protein prenylation via the MVA pathway might be an important target for CSC suppression. In addition, Freed-Pastor and Mizuno et al. reported that mutant p53 upregulated expression of MVA pathway enzymes [17]. The tumor characteristics of metabolic subtypes of CRC, such as consensus molecular subtype 3 or KRAS mutations, may also be related to MVA pathway activation.

Metformin is a well-known AMPK activator, and activation of AMPK inhibits mTOR signaling and energy-consuming pathways [26,27]. Ching and Abraham et al. reported that activation of AMPK inhibits acetyl-CoA carboxylase (ACC) activity, which is related to fatty acid synthesis [28]. Many studies have reported that metformin induces anticancer activity by AMPK activation and mTOR inhibition. Thus, we also investigated the relationship between AMPK/mTOR regulators and the MVA pathway, and confirmed that an AMPK activator and an mTOR inhibitor reduced CSC populations; these effects were reversed by adding mevalonate. Moreover, the AMPK activator and mTOR inhibitor decreased the expression of mediator enzymes of the MVA pathway, suggesting regulation of this pathway by AMPK/mTOR signaling. In addition, we found that metformin-resistant cells (SW620) did not show significant change of AMPK/mTOR signals and reduction of key enzymes expression of the MVA pathway by treatment of metformin (Appendix A).

Statin is a drug that lowers cholesterol through HMGCR inhibition and can decrease cancer risk [29]. We also showed that statin was a potent inhibitor of HMGCR and had a CSC-suppressive effect. However, the inhibitory effects of statin on key enzymes of the MVA pathway were weaker and less consistent than those of metformin, suggesting the importance of additional effects of metformin on the MVA pathway via AMPK activation and mTOR inhibition. Based on the mechanism of CSC suppression by inhibition of protein prenylation, these candidate drugs, such as metformin and AMPK activators, could be useful for adjunctive treatment in chemotherapy for advanced CRC or in chemoprevention for high-risk CRC groups if they have a wide margin of safety.

In summary, we investigated the detailed relationship between metformin-induced suppression of CSCs and the MVA pathway. Metformin-induced CSC suppression could occur through AMPK activation and protein prenylation inhibition, rather than through inhibition of cholesterol synthesis via the MVA pathway (Figure 5). However, because metformin has many molecular mechanisms of antitumor effect, we could not elucidate the detailed interaction between prenylation-dependent and other direct and indirect mechanisms of metformin-induced antitumor or CSC suppression. Further studies would be needed to investigate these cross-reactions among direct and indirect mechanisms of the CSC-suppressing effect of metformin.

## 4. Methods

### 4.1. Cell Lines and Culture Conditions

HT29 and DLD-1 colorectal cancer cell lines were purchased from the American Type Culture Collection (Manassas, VA, USA). Cell lines were maintained in Dulbecco modified Eagle’s medium (DMEM; Invitrogen, Carlsbad, CA, USA) supplemented with 10% fetal bovine serum (Gibco, Franklin Lakes, NJ, USA) and 1% penicillin/streptomycin (Invitrogen, Carlsbad, CA, USA) at 37 °C in 5% CO_2_.

### 4.2. Tumor Sphere Culture Assay

HT29 and DLD-1 cells (2000 or 4000 cells per well) were plated in 24-well ultra-low adhesive plates (Corning Incorporated, Corning, NY, USA) in sphere formation medium with drugs for 5 or 7 days. The sphere formation medium was serum-free DMEM-F12 supplemented with B27 (Life Technologies, Carlsbad, CA, USA), 20 ng/mL epidermal growth factor, 10 ng/mL basic fibroblast growth factor (R & D Systems, Minneapolis, MN, USA), 1% penicillin/streptomycin, and 2 mM L-glutamine (Life Technologies, Carlsbad, CA, USA). Cells were incubated in a 5% CO_2_ chamber at 37 °C, and 500 μL of the culture medium was changed every 48 h. After 5 or 7 days, the number of tumor spheres was counted under a microscope (Olympus, Tokyo, Japan, BX51 microscope).

### 4.3. Microarray Data Analysis Using a Public Data Set

We performed a microarray data analysis of GSE76342 using the limma package in R. Limma is an R/Bioconductor software package that provides an integrated solution for analyzing data from gene expression experiments. CEL files have been deposited to the Gene Expression Omnibus database (http://www.ncbi.nlm.nih.gov/geo) under the accession number GSE76342.

### 4.4. Flow-Cytometric Analysis and Fluorescence-Activated Cell Sorting (FACS)

HT29 and DLD-1 were plated in 6-well plates at a density of 2 × 10^5^ cells/well and treated with metformin and other reagents. After 48 h, the prepared cells were detached with Accutase (Millipore, Billerica, MA, USA) and resuspended in FACS buffer (1 × PBS, 1% bovine serum albumin, and 2 mM ethylenediaminetetraacetic acid). Primary antibodies against CSC markers (PE-conjugated anti-CD133 and FITC-conjugated anti-CD44) were added. Specific information of antibodies used in the study was given in Appendix A. Samples were incubated for 10 min at 4 °C, washed with FACS buffer, and subjected to flow cytometry for analysis using a FACSVerse (BD Biosciences, San Diego, CA, USA) coupled to a computer with BD FACSuite software.

### 4.5. RNA Extraction and qPCR

Total RNA was isolated using TRIZOL Reagent (Invitrogen). Equal amounts of cDNA were synthesized using the reverse transcription 5× Master Pro Mix (ELPISBIO, Daejeon, Korea), and mixed with 2× SYBR Green with high ROX (ELPISBIO). qPCR reactions were performed using the gene-specific primers in Appendix A. All qPCRs performed using SYBR Green were conducted at 55 °C for 10 min, 95 °C for 10 min, and 40 cycles of 95 °C for 15 s and 60 °C for 1 min. The specificity of the reaction was verified by melt curve analysis.

### 4.6. Western Blotting

Prepared cells were lysed using a protein extraction solution (iNtRON Biotechnology, Gyeonggi, Korea). After protein quantification, 20 μg portions of protein extracts were fractionated using 12% or 15% sodium dodecyl sulfate polyacrylamide gel electrophoresis and transferred to polyvinylidene fluoride membranes (Bio-Rad, Hercules, CA, USA). After blocking with 10% skim milk, membranes were incubated with primary antibodies overnight at 4 °C. Subsequently, membranes were incubated with secondary antibodies for 1 h at room temperature. Proteins were detected using an ECL Western blotting detection kit (Amersham Biosciences, Freiburg, Germany) and light was captured on Kodak image film.

For Western blotting for prenylated proteins, prepared cells were harvested and lysed in 0.2 mL of high concentration of salt buffer (50 mM Tris-HCl, pH 7.5/150 or 450 mM NaCl, 0.5% NP-40). Then, protein lysate was resolved in 12% or 15% SDS/polyacrylamide gel.

### 4.7. Drugs and Antibodies

Metformin and FTI-277 were purchased from Sigma (St. Louis, MO, USA). AICAR, simvastatin, GGTI-298, and YM-53601 were purchased from Cell Signaling Technology (Danvers, MA, USA), Merck Millipore (Darmstadt, Germany), R & D systems (Minneapolis, MN, USA), Cayman (Ann Arbor, Michigan, AN, USA). Mevalonate, rapamycin were purchased from Santa Cruz Biotechnology (Santa Cruz, CA, USA). The antibodies used for flow cytometry, Western blotting, and IHC analysis included: anti-AMPK, anti-pAMPK (Thr172), anti-S6, and anti-p-S6 (Ser235/236) (Cell Signaling Technology, Danvers, MA, SAD), anti-GGPS1 (Santa Cruz Biotechnology, Dallas, TX, USA), anti-FDPS (Abcam, Cambridge, UK), anti-PROM1 (CD133) (MACS, Bergisch Gladbach, Germany), anti-CD44 (eBioscience, San Diego, CA, USA), phycoerythrin (PE)-conjugated anti-PROM1 (CD133), and fluorescein (FITC)-conjugated anti-CD44 (MACS). Additional information on antibodies and drugs is shown in Appendix A.

### 4.8. In Vivo Mouse Xenograft Experiments

Six-week-old male BALB/c athymic nude mice were purchased from Central Lab (Seoul, Korea) and acclimated for 1 week. Mouse experiments were performed in accordance with protocols approved by the Committee on Care and Use of Laboratory Animals of Yonsei University College of Medicine (Seoul, Korea) and according to institutional guidelines and policies.

HT29 cells were suspended in Matrigel (BD Bioscience) at a density of 1 × 10^6^ cells/200 μL, diluted 1:1 in PBS, and subcutaneously injected into both flanks of the mice. After 2 weeks, the implanted mice were randomly divided into four groups (5 mice per group): control, metformin only, mevalonate only, and combination (metformin and mevalonate). All drugs were injected intraperitoneally (metformin: 250 mg/kg in 200 μL PBS; mevalonate: 10 mg/kg in 200 μL PBS) on a daily basis for 21 days. Vehicle control (200 μL of PBS) was injected intraperitoneally into the mice in the control, metformin-only, and mevalonate-only groups. Tumor sizes were measured each day using calipers, and tumor volumes were calculated based on the following formula: tumor volume = length × (width)^2^ / 2. All mice were sacrificed 21 days after the first drug treatment and the tumor masses were dissected. The dissected tumors were placed in 4% paraformaldehyde (PFA) for immunohistochemistry (IHC) and were collected for further analysis.

### 4.9. Immunohistochemistry

IHC for CD133, CD44, GGPS1, and FDPS was performed on 4 μm sections of formalin-fixed, paraffin-embedded, dissected tumor samples. The paraffin-embedded sections were deparaffinized in xylene and rehydrated in gradually decreasing concentrations of ethanol. Antigen retrieval was performed using sodium citrate buffer (10 mM, pH 6.0) in a heated pressure cooker for 5 or 7 min. After incubation with 3% hydrogen peroxide for 30 min to block endogenous peroxidase activity, sections were incubated in a blocking reagent for 30 min at room temperature. Sections were incubated with primary antibodies overnight at 4 °C, followed by secondary antibody for 30 min at room temperature. After slides were developed with a Vectastain ABC kit (Vector Laboratories, Burlingame, CA, USA), immunodetection was performed using DAB solution (Dako, Carpinteria, CA, USA). After counterstaining with hematoxylin, IHC staining was evaluated by light microscopy and immunoactivity was assessed based on the proportion of immunostained tumor cells. Additional information on antibodies is shown in Appendix A.

We measured the intensity of CSC markers (CD44 and CD133) and IHC scores using IHC profiles based on the Image J program [30]. The IHC score calculation was score = (number of pixels in a zone × score of the zone)/total number of pixels in the image. The score of the zone was assigned as 4 for high positive zones, 3 for positive zones, 2 for low positive zones, and 1 for negative zones.

### 4.10. Statistical Analysis

Statistical analyses were performed using IBM SPSS Statistics version 20.0 (IBM Co., Armonk, NY, USA). For the evaluation of two data sets, unpaired Student’s *t* tests or Mann–Whitney tests were performed. To evaluate more than two groups, one-way or two-way analysis of variance was applied. Every experiment was conducted at least in triplicate to ensure reliability. All calculated *p*-values were two-sided and *p* < 0.05 was considered statistically significant.

## 5. Conclusions

We investigated mechanism of metformin-induced CSC-inhibitory effect in the field of tumor metabolism, and demonstrated that the CSC-suppressive effect of metformin was associated with AMPK activation/mTOR inhibition and repression of protein prenylation through suppression of mevalonate pathway in colorectal cancer. In the future, clinical usefulness of metformin might be considered as chemopreventive or adjunctive treatment for colorectal tumor.

## Figures and Tables

**Figure 1 cancers-12-02554-f001:**
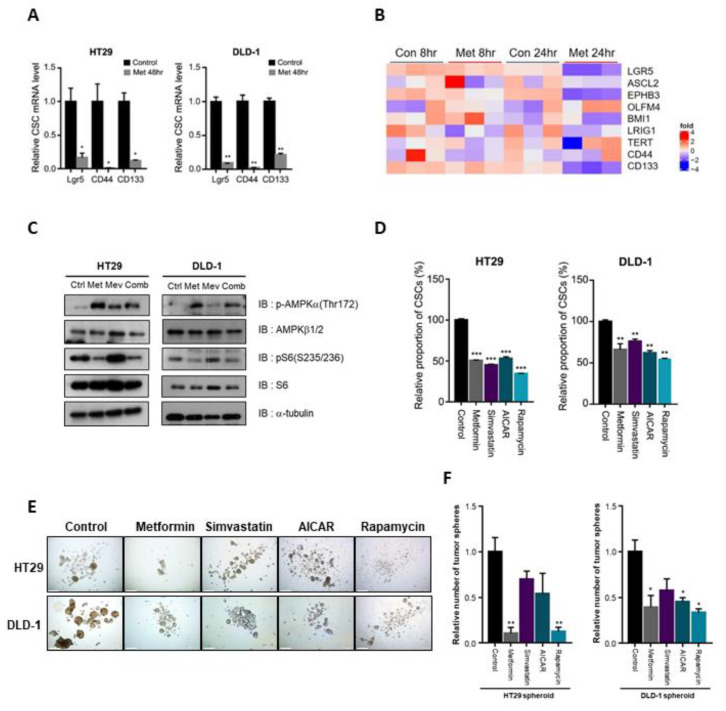
Suppression of cancer stem cells by metformin, statin, AMPK activator, and mTOR inhibitor. (**A**) After 48 h of treatment with metformin (10 mM), mRNA levels of CSC markers in HT29 and DLD-1 cells were analyzed by qPCR. (**B**) Heatmap analysis representing transcripts of CSC markers using public RNA-sequencing data of LoVo cells treated with and without metformin for 8 h and 24 h (GSE76342). (**C**) Western blot analysis of p-AMPK and p-S6 expression levels in HT29 and DLD-1 cells after 48 h of treatment with control vehicle, metformin (10 mM), mevalonate (1 mM), or combination (metformin (10 mM) and mevalonate (1 mM)). (**D**) HT29 and DLD-1 cells were treated with metformin (10 mM), simvastatin (2 μM), AICAR (1 mM), or rapamycin (200 nM). After 48 h, the proportions of CD44+/CD133+ cells were analyzed by flow cytometry using anti-CD44-FITC and anti-CD133-PE. (**E**,**F**) In tumor sphere cultures treated with metformin (10 mM), simvastatin (2 μM), AICAR (1 mM), or rapamycin (200 nM) for 7 days, the number of tumor spheres (≥200 µm in diameter) was counted and compared to the counts in control cultures. Data are expressed as the mean ± standard error of three independent experiments; * *p* < 0.05, ** *p* < 0.01, and *** *p* < 0.001 (compared with control).

**Figure 2 cancers-12-02554-f002:**
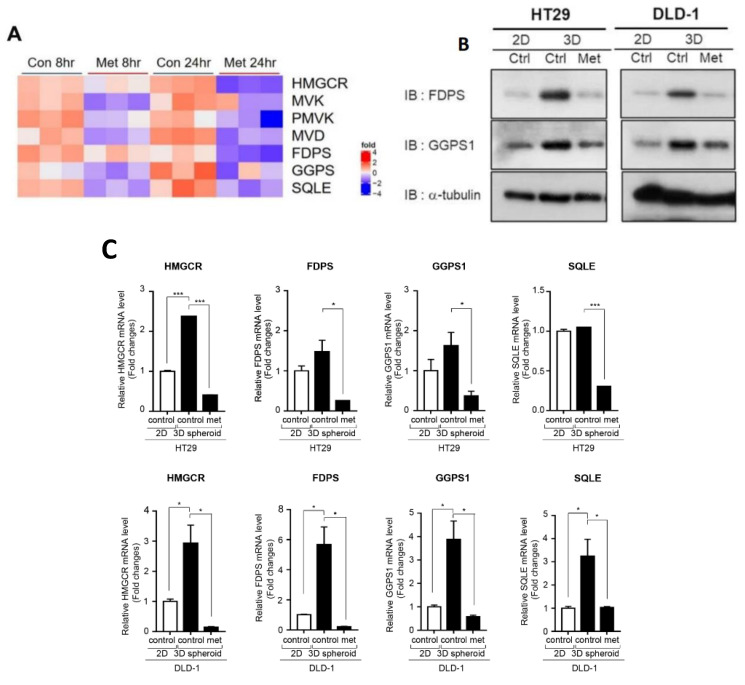
Regulation of enzymes of the mevalonate pathway and cancer stem cell population by metformin, AMPK activator, mTOR inhibitor, and mevalonate. (**A**) Heatmap analysis representing transcripts of enzymes of the MVA pathway using the same public RNA-sequencing data set as in Figure 1B. (**B**) Western blot analysis for the expression of FDPS and GGPS1 was performed after 5 days of 2D adherent culture and 3D tumor sphere culture and after 5 days’ treatment with metformin (10 mM). (**C**) mRNA levels of HMGCR, FDPS, GGPS1, and SQLE were measured by qPCR under the same conditions as in (**B**). (**D**) HT29 and DLD-1 tumor spheres were treated with metformin (10 mM), simvastatin (2 μM), AICAR (1 mM), rapamycin (200 nM), or mevalonate (1 mM) for 7 days and then subjected to qPCR analysis of HMGCR, FDPS, GGPS1, and SQLE. (**E**) HT29 and DLD-1 cells were treated with metformin (10 mM), simvastatin (2 μM), AICAR (1 mM), rapamycin (200 nM), mevalonate (1 mM), or a combination of each drug with mevalonate (1 mM) for 48 h. Cells were subjected to flow cytometry to analyze CD44+/CD133+ cells using anti-CD44-FITC and anti-CD133-PE. (**F**) In tumor sphere cultures with metformin (5 mM), mevalonate (1 mM), or a combination of metformin (5 mM) and mevalonate (1 mM) for 7 days, the number of tumor spheres (≥200 µm in diameter) was counted and compared to the counts in control cultures. Magnification, 40×. Data are expressed as the mean ± standard error of three independent experiments; * *p* < 0.05, ** *p* < 0.01, and *** *p* < 0.001 (compared with control).

**Figure 3 cancers-12-02554-f003:**
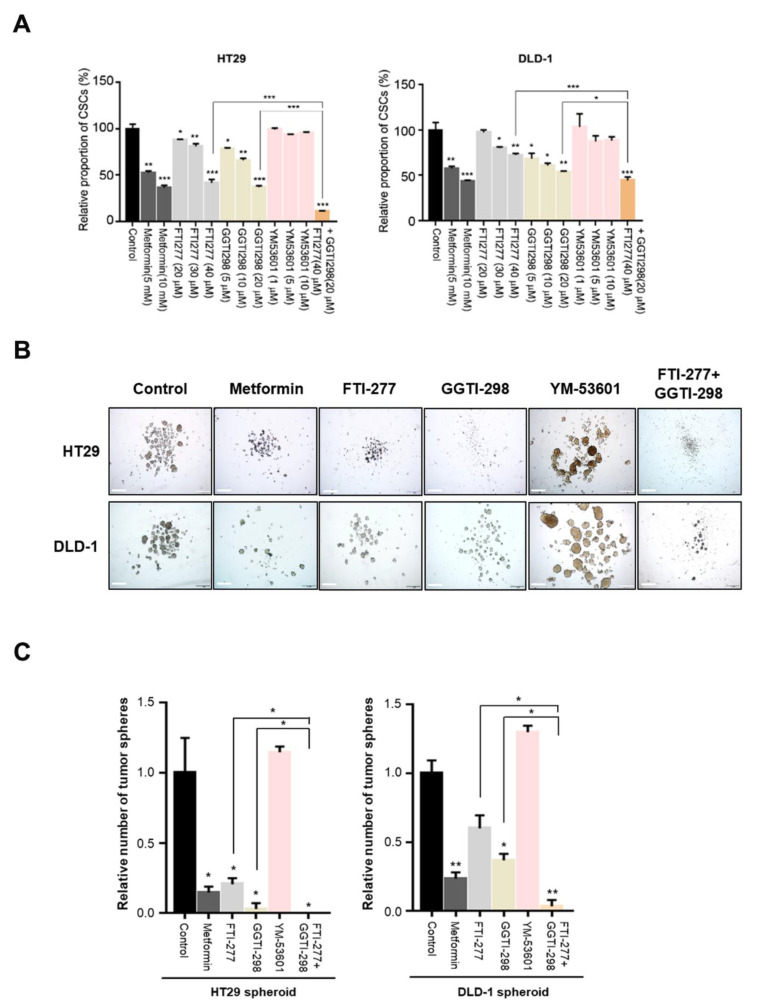
Cancer stem cell proportion was reduced by inhibition of protein prenylation, but not by inhibition of squalene synthase in the mevalonate pathway. (**A**) HT29 and DLD-1 cells were treated with metformin (5 or 10 mM), farnesyltransferase inhibitor (FTI-277; 20, 30, or 40 μM), geranylgeranyl transferase I inhibitor (GGTI-298; 5, 10, or 20 μM), or squalene synthase inhibitor (YM-53601; 1, 5, or 10 μM) for 48 h, and then the CD44+/CD133+ population was analyzed by flow cytometry using anti-CD44-FITC and anti-CD133-PE. (**B**,**C**) In tumor sphere cultures treated with metformin (10 mM), FTI-277 (40 μM), GGTI-298 (20 μM), or YM-53601 (10 μM) for 5 days, the number of tumor spheres (≥200 µm in diameter) was counted and compared to the counts in control cultures. Magnification, 40×. Data are expressed as the mean ± standard error of three independent experiments; * *p* < 0.05, ** *p* < 0.01, and *** *p* < 0.001 (compared with control).

**Figure 4 cancers-12-02554-f004:**
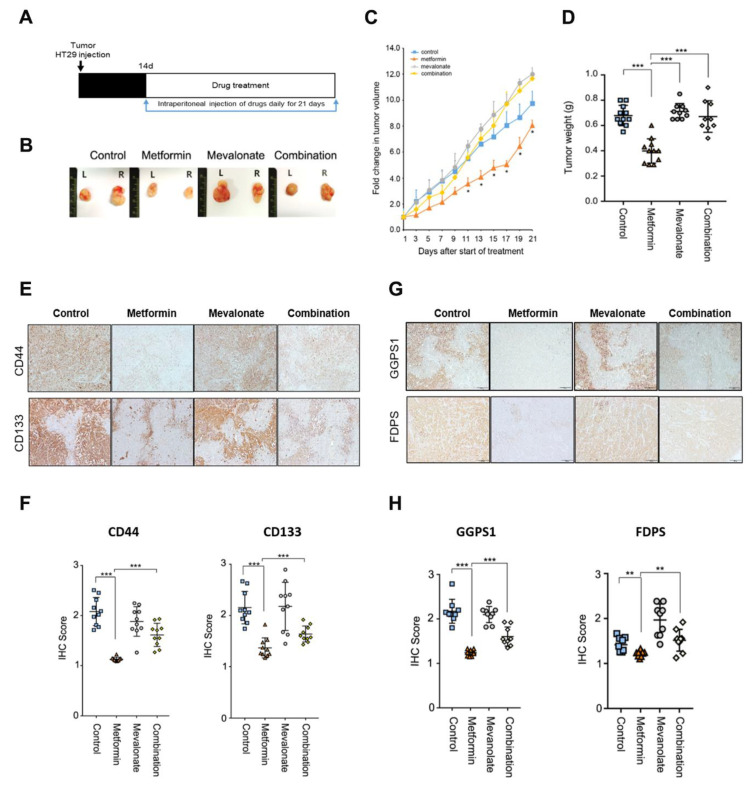
The tumor-suppressing effect of metformin was reversed by addition of mevalonate in a mouse xenograft model. (**A**,**B**) HT29 cells were injected subcutaneously into both flanks of nude mice. Implanted mice were randomly and evenly allocated into four groups: vehicle, metformin (by intraperitoneal injection, 250 mg/kg in 200 μL PBS daily), mevalonate (by intraperitoneal injection, 10 mg/kg in 200 μL PBS daily), or the combination of the same daily doses of both metformin and mevalonate. Tumors were measured, and mice were sacrificed on day 21. In mice implanted with HT29 cells, tumor growth (**C**) and tumor weights (**D**) were compared between the four groups. In (**C**), * denotes *p* < 0.05 for comparison with the combination group. (**E**) IHC to evaluate expression of CD44 and CD133 was performed on sections of formalin-fixed, paraffin-embedded, dissected xenograft tumor samples from mice treated with vehicle, metformin alone, mevalonate alone, or metformin combined with mevalonate. (**F**) IHC scores for CD44 or CD133-stained cells were analyzed using Image J. (**G**) IHC was performed for FDPS and GGPS1 and (**H**) analyzed using Image J. Magnification, 200×. Data are expressed as the mean ± standard error; *n* = 10 per group. * *p* < 0.05, ** *p* < 0.01, and *** *p* < 0.001.

**Figure 5 cancers-12-02554-f005:**
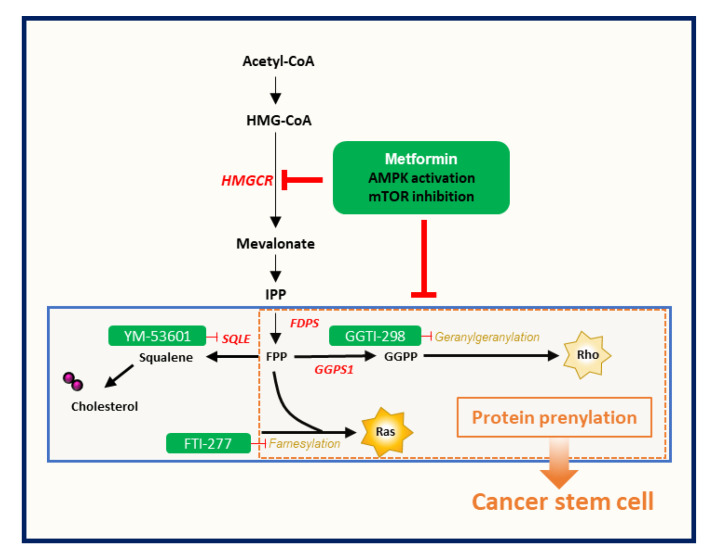
Metformin-induced CSC suppression through AMPK activation and inhibition of protein prenylation by suppression of the mevalonate pathway.

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
