# Peer review of "Metformin Suppresses Cancer Stem Cells through AMPK Activation and Inhibition of Protein Prenylation of the Mevalonate Pathway in Colorectal Cancer"

_cancers, 2020, doi:10.3390/cancers12092554_

Round 1
Reviewer 1 Report

Remarks to the Authors:
The work was aimed to uncover the relationship between metformin-induced suppression of CSCs and the MVA pathway in colorectal cancer. 
There are several weaknesses, a few major flaws and some missing critical experiments that prevent this manuscript from being recommended for publication. Importantly, the manuscript lacks the novelty and it is not very innovative. All conclusions stated by the authors were already being discussed in several other articles, readily available in the literature. Furthermore, the discussion is not well balanced and adequately supported by the data. Although the article is quite well presented, the quality is not adequate to this journal and the authors are discouraged to consider a resubmission in the future because the article lacks scientifically sound.
Major Concerns:
- Authors recently published that the inhibition of the glutamine pathway could enhance the CSC-suppressing effect of metformin in CSCs of each CRC cell line (HT29 and SW620) (Kim JH et al, 2018 Scientific Reports). The authors should focus their attention on the balance and relationship between glutamine metabolism and mevalonate pathway after metformin treatment in CSC.
- To validate their findings the analysis should be controlled by a metformin-resistant cell model and CSC cell lines with the different genetic backgrounds to exclude the involvement, for example, of p53, RAS/RAF and/or other players in repression/activation of mevalonate pathway.
- The authors should investigate and exclude if the effect depend on a direct player, Foxo3A transcription factors (AMPK-FoxO axis), involved in important metabolic function in cancer cells. Recent evidence showed that in metabolically stressed cancer cells and tumors, FoxO3A is recruited to the mitochondrial surface in a MEK/ERK- and AMPK-dependent manner, while only the MEK/ERK signal is required in cancer cells treated with chemotherapeutic agents. After cleavage of its N- terminal domain, FoxO3A is imported into the mitochondrial matrix where it activates a transcriptional program leading to cancer cell survival. On the other hand, mitochondrial FoxO3A is required for apoptosis induction in cancer cells treated with metformin.
- The authors should describe better the methods used for tumorspheres counting (i.e. number and size).
- Figure 2 C,D should be changed as fold induction.
- Figure 3 suffers from a lack of controls. Moreover, the authors should validate the findings on protein prenylation with specific assay available in vitro and in vivo.
Overall rating
The manuscript is not recommended for publication.
Author Response
Response to Reviewer 1 Comments
Referee #1 :
The work was aimed to uncover the relationship between metformin-induced suppression of CSCs and the MVA pathway in colorectal cancer. There are several weaknesses, a few major flaws and some missing critical experiments that prevent this manuscript from being recommended for publication. Importantly, the manuscript lacks the novelty and it is not very innovative. All conclusions stated by the authors were already being discussed in several other articles, readily available in the literature. Furthermore, the discussion is not well balanced and adequately supported by the data. Although the article is quite well presented, the quality is not adequate to this journal and the authors are discouraged to consider a resubmission in the future because the article lacks scientifically sound.
We appreciate your constructive comments. We understand your concerns. As you suggested, we have re-written or revised the result portion. Meanwhile, we have also cautiously exerted our best efforts to clarify our opinions so that they may be received. Please review the output in a positive light.
Major Concerns:
Point 1. Authors recently published that the inhibition of the glutamine pathway could enhance the CSC-suppressing effect of metformin in CSCs of each CRC cell line (HT29 and SW620) (Kim JH et al, 2018 Scientific Reports). The authors should focus their attention on the balance and relationship between glutamine metabolism and mevalonate pathway after metformin treatment in CSC.
Response 1: Thank you for your comment. We understand the concern. As you know, we recently showed the difference of glutamine metabolism between metformin-sensitive and –resistant CRC cell lines. In other words, we demonstrated the mechanism of metformin-resistance using metformin-resistant cell lines. However, in the current study, we focused on the action mechanism of metformin in metformin-sensitive cell lines.
It is known that metformin acts as HMG-CoA reductase inhibitor, AMPK activator, and mTOR inhibitor. However, their direct relationship with suppression of cancer stem cell, especially through inhibition of mevalonate pathway and protein prenylation, might be a novel finding in our current study.
Point 2. To validate their findings, the analysis should be controlled by a metformin-resistant cell model and CSC cell lines with the different genetic backgrounds to exclude the involvement, for example, of p53, RAS/RAF and/or other players in repression/activation of mevalonate pathway.
Response 2: We appreciate your pointing out this issue. We agree that it will be better to add the experiment that you suggested.
However, it would be difficult to identify the direct relationship between genetic backgrounds and mevalonate pathway only with experiments using several cell lines with different genetic background, and this issue is beyond the scope of our study.
Therefore, we added the results focusing on metformin-resistant cell line (SW620). As a result, metformin treatment induced no significant changes in expressions of p-AMPK and p-S6 in WB, and key enzymes of MVA pathways in qPCR. We added these results in Supplementary Fig 3. and explanation in Discussion section (line 278-280), as follows.
“In addition, we found that metformin-resistant cells (SW620) did not show significant change of AMPK/mTOR signals and reduction of key enzymes expression of the MVA pathway by treatment of metformin (Supplementary Figure 3)."
Figure legend:
Supplementary Figure 3. Using metformin-resistant cell line, SW620, Western blot analysis (p-AMPK/p-S6 and FDPS/GGPS1) and qPCR analysis (HMGCR, FDPS, GPS1 and SQLE) were done in the same conditions as in Fig 1(C) and Fig 2(D), respectively.
Point 3. The authors should investigate and exclude if the effect depends on a direct player, Foxo3A transcription factors (AMPK-FoxO axis), involved in important metabolic function in cancer cells. Recent evidence showed that in metabolically stressed cancer cells and tumors, FoxO3A is recruited to the mitochondrial surface in a MEK/ERK- and AMPK-dependent manner, while only the MEK/ERK signal is required in cancer cells treated with chemotherapeutic agents. After cleavage of its N- terminal domain, FoxO3A is imported into the mitochondrial matrix where it activates a transcriptional program leading to cancer cell survival. On the other hand, mitochondrial FoxO3A is required for apoptosis induction in cancer cells treated with metformin.
Response 3: Thank you for your interesting comments. Metformin is a well-known AMPK activator, and has been known to be related with various pathways associated with anti-tumor effect. As you mentioned, AMPK-FoXO3A axis is an important pathway related with metformin-induced apoptosis. However, the effect of metformin on FOXO3a might be dependent on conditions such as cell lines, because the activation of p-AMPK to metformin is consistent, but FOXO3a response to metformin is different, as shown in the following references.
At any rate, we do not know the effect of FOXO3a on cancer stem cell, and it is possible that FOXO3a could be one of the action mechanisms of metformin-induced suppression of cancer stem cells. However, our point of view is different, and we focused on the effect of metformin on direct relationship between cancer stem cell and mevalonate pathway (especially, protein prenylation). Therefore, this issue is also beyond the scope of our study.
Then, we added sentences in the last paragraph of Discussion section (line 292 –296) about the necessity to elucidate the detailed interaction between prenylation-dependent and other direct and indirect mechanisms of metformin-induced anti-tumor or CSC suppression, as follows.
“However, because metformin has many molecular mechanisms of anti-tumor effect, we could not elucidate the detailed interaction between prenylation-dependent and other direct and indirect mechanisms of metformin-induced anti-tumor or CSC suppression. Further studies would be needed to investigate these cross-reactions among direct and indirect mechanisms of CSC suppressing effect of metformin.”
References:
- Metformin suppresses expression of the selenoprotein P gene via an AMP-activated kinase (AMPK)/FoxO3a pathway in H4IIEC3 hepatocytes. Takayama H, et al. J Biol Chem. 2014 Jan 3;289(1):335-45.
- Metformin induces apoptosis and cell cycle arrest mediated by oxidative stress, AMPK and FOXO3a in MCF-7 breast cancer cells. Queiroz EA, et al. PLoS One. 2014 May 23;9(5):e98207.
- Metformin reduces the Walker-256 tumor development in obese-MSG rats via AMPK and FOXO3a. de Queiroz EA, et al. Life Sci. 2015 Jan 15;121:78-87.
- Activation of AMPK inhibits cervical cancer cell growth through AKT/FOXO3a/FOXM1 signaling cascade. Yung MM, et al. BMC Cancer. 2013 Jul 3;13:327.
Point 4. The authors should describe better the methods used for tumorspheres counting (i.e. number and size).
Response 4: Thank you for your comment. We counted significantly larger tumorspheres because smaller tumorspheres were debris or dying without change in size. Therefore, we counted the number of tumorsphere based on the diameter of over 200 µm. We added this description in Figure legends (Figure 1, 2, 3) as follows: the number of tumor spheres (≥ 200 µm in diameter) was counted.
Point 5. Figure 2 C, D should be changed as fold induction.
Response 5: In light of your comment, we have modified the name of Y axis ‘Relative mRNA levels (Fold changes)’ in Figure 2 C, D.
Point 6. Figure 3 suffers from a lack of controls. Moreover, the authors should validate the findings on protein prenylation with specific assay available in vitro and in vivo.
Response 6: Thank you for your important comment. There are controls in experiments of Figure 3. For validation of metformin-induced suppression of protein prenylation, we performed WB analyses of KRAS for farnesylation and RalA for geranylgeranylation after treatment of metformin, and identified the change of shifted bands by farnesylation and geranylgeranylation. Then, we found that metformin treatment suppressed the shifted bands in WB analyses. Because it was difficult to separate the prenylated protein bands, the shifted bands were weak or not enough. We added this result in Supplementary Figure 1, and explained it in Results section (line 173-177), as follow.
“In addition, to confirm the decrease of protein prenylation by metformin, we performed Western blot analysis to identify the change of shifted prenylated protein bands of RAS and Ral A by farnesylation and geranylgeranylation, respectively. Although the shifted bands were weak in some conditions, we found the decrease of shifted prenylated protein bands of RAS and Ral A (Supplementary Figure 1).
Figure legend:
Supplementary Figure 1. After 5 days’ treatment with metformin (10 mM), tumor spheres were harvested and lysed in high concentration of salt buffer (50 mM Tris-HCl, pH 7.5/ 450 mM NaCl, 0.5% NP-40). Then, protein lysate was resolved in 15% SDS/polyacryl-amide gel, and Western blot analysis was performed to identify shifted prenylated protein bands (arrow head) of RAS (A) and Ral A (B).

Reviewer 2 Report
Figure 5. In the flow diagram indicating the inhibitors used in the study will help to visualize the authors conclusion of the study.
Author Response
Response to Reviewer 2 Comments
Referee #2 :
Point 1 : Figure 5. In the flow diagram indicating the inhibitors used in the study will help to visualize the authors conclusion of the study.
Response 1: Thank you for your comment. We added the name of inhibitors used in diagram.

Reviewer 3 Report
This paper illustrates a potential biochemical explanation for the anti-CSCs activity of metformin in colorectal cancer. Overall the data are clearly presented and the quality of the experiment is good. However, a number of issues arise from this work, listed below:
-The graph in fig1A and the heatmap in fig1B and 2A all show how metformin treatment decreases the mRNA levels of several stemness or biosynthetic genes. Can the authors provide one or more genes that are on the other hand upregulated after metformin treatment in order to rule out a potential widespread transcriptional inhibitory effect of this drug?
-Do simvastatin, AICAR and rapamycin affect the signalling that is shown in figure 1C? Is there a possible explanation why the effect of CSCs is so striking while the effect on the number of spheres is not? Is it a matter of concentrations used?
-I don't understand why in figure 2 the cells grown in 2D are only used as control and not also tested after metformin treatment. Even if the effect shown was restricted to the 3D grown cell spheres, I consider this result relevant anyway.
-In line 142-143 the authors write that the increased expression of biosynthetic enzymes suggest that MVA pathway may have a role in tumorigenesis by promoting CSCs. Does MVA also increase the expression of stemness genes? This would reinforce their claim.
-The antitumor effect of metformin is clearly lost by coadministration with mevalonate in the xenograft model. However, this anticancer effect can be associated to many different biologic activities of AMPK. Ideally, replicating the same results with FFTi or GGTi or the combinations used in fig3 would be desirable. Moreover, even if metformin decreases CSC markers in xenograft tumors, there is no direct evidence that the growth delay is due to CSC depletion or simply to an antiproliferative effect on cancer cells. Does metformin have an antiproliferative effect at the dose used in this study?
-In the discussion, the authors speculate on the mechanisms linking prenylation to CSC generations through the indirect effect on RAS signaling. However, one the models used in this study (HT29 cell line) is BRAF mutant. What is the possible mechanism of action in this case? Rho signaling can still be responsible in this case, but no results about this are shown. Adding some results on the possible mechanism linking the inhibition of prenylated proteins to the decrease of CSCs would really increase the relevance of the paper.
-Another relevant point in the discussion is not clear: do the Authors evisage a potential clinical applicability of metformin as a CSC inhibitor? This leads to a more fundamental question: are the concentrations of metformin used in this study clinically relevant and achievable in patients? Moreover, statin effect was not shown to be relevant as single agent in CSCs, but no combination was obtained with metformin. Probably this goes beyond the scope of the paper, but such combination could be a possible approach.
-Is there a role for FFTi or GGTi in cancer treatment? Are there any supporting clinical trials ongoing or with results?
Finally, I believe there is a problem with the references, since the first to appear is number 8 (line 45). Moreover, after each reference numer follows the author of the referenced article, as if something went wrong with the bibliography manager. Probably, the reference list was set in alphabetical order, but this is not the right format for scientific articles on this journal.
Author Response
Response to Reviewer 3 Comments
Referee # 3 :
This paper illustrates a potential biochemical explanation for the anti-CSCs activity of metformin in colorectal cancer. Overall the data are clearly presented and the quality of the experiment is good. However, a number of issues arise from this work, listed below:
Point 1. The graph in fig1A and the heatmap in fig1B and 2A all show how metformin treatment decreases the mRNA levels of several stemness or biosynthetic genes. Can the authors provide one or more genes that are on the other hand upregulated after metformin treatment in order to rule out a potential widespread transcriptional inhibitory effect of this drug?
Response 1: Thank you for your comments.
The heatmap data was produced using public data (GSE76342); He, J., Wang, K., Zheng, N. et al. Sci Rep 5, 17423 (2015). He, J., et al. also showed 983 genes up-regulated by metformin, which are related with various cell activities, cellular metabolic processes, cell cycle, and regulation of transcription, and immunity.
Point 2. Do simvastatin, AICAR and rapamycin affect the signalling that is shown in figure 1C? Is there a possible explanation why the effect of CSCs is so striking while the effect on the number of spheres is not? Is it a matter of concentrations used?
Response 2: Thank you for your comment. We found no change of pAMPK and pS6 by simvastatin treatment, and AICAR-induced change of p-AMPK/p-S6 and rapamycin-induced suppression of p-S6 were found as shown in metformin treatment, as follows. (image attached word file)
As for the difference of drug effect between CSC flow cytometry analysis and sphere formation, there would be several factors related with this difference, such as characteristics of cell lines, drug concentration, treatment duration, and action mechanisms of drugs, which could influence via indirect or unknown pathway related with sphere formation.
Point 3. I don't understand why in figure 2 the cells grown in 2D are only used as control and not also tested after metformin treatment. Even if the effect shown was restricted to the 3D grown cell spheres, I consider this result relevant anyway.
Response 3: Thank you for your comment. We also performed the experiments in 2D culture, but the results were not consistent, compared to 3D. As you know, the results of experiments in 3D sphere culture would be more specific to cancer stem cells because there exist high percentage of CSCs in 3D sphere culture.
Point 4. In line 142-143 the authors write that the increased expression of biosynthetic enzymes suggest that MVA pathway may have a role in tumorigenesis by promoting CSCs. Does MVA also increase the expression of stemness genes? This would reinforce their claim.
Response 4: Thank you for your important comment. We already showed increased population of CSCs by treatment of mevalonate in flow cytometry analysis using CSCs markers (CD133, CD44) in Figure 2E. In addition, we performed additional experiment of qPCR for more CSC markers after treatment of mevalonate, and found most mRNA levels of CSC markers were increased by mevalonate in both cell lines as follows. (image attached word file)
Point 5. The antitumor effect of metformin is clearly lost by coadministration with mevalonate in the xenograft model. However, this anticancer effect can be associated to many different biologic activities of AMPK. Ideally, replicating the same results with FFTi or GGTi or the combinations used in fig3 would be desirable. Moreover, even if metformin decreases CSC markers in xenograft tumors, there is no direct evidence that the growth delay is due to CSC depletion or simply to an antiproliferative effect on cancer cells. Does metformin have an antiproliferative effect at the dose used in this study?
Response 5 : Thank you for your comments. Even if treatment of FFTi and/or GGTi would show the same results with those by metformin treatment in in vivo model, it could not explain the exact mechanism via inhibition of protein prenylation by metformin. To confirm exactly the direct anti-tumor effect of metformin via inhibition of protein prenylation using in vivo model, in addition to your comment, it would be necessary to show that overexpressed enzymes of protein prenylation could reverse metformin-induced suppression of tumor. However, currently it would be difficult to satisfy these experimental conditions.
According to your comments, we performed Ki67 staining to show anti-proliferative effect of metformin, and found a significant decrease of Ki67 staining by treatment of metformin in xenograft model (Supplementary Fig 2). Therefore, tumor-suppressing effect of metformin would not depend on only depletion of CSCs. However, because the CSCs are important for tumor initiation, persistent progression and metastasis, and the effect of metformin was more prominent on CSC population in our previous data [8], we described it with a focus on CSC depletion by metformin. We added this comment in Results section (line 210-215), as follows.
“In addition, we performed Ki67 staining to show anti-proliferative effect of metformin, and found a significant decrease of Ki67 staining by treatment of metformin (Supplementary Fig 2). Therefore, tumor-suppressing effect of metformin would not depend on only suppression of CSCs. However, because the CSCs are important for tumor initiation, persistent progression and metastasis, and the effect of metformin was more prominent on CSC population in our previous data [8], we focused on CSC suppression by metformin.”
Figure legend:
Supplementary Figure 2. IHC to evaluate expression of Ki67 was performed on sections of formalin-fixed, paraffin-embedded, dissected xenograft tumor samples from mice treated with vehicle, metformin alone, mevalonate alone, or metformin combined with mevalonate.
Point 6. In the discussion, the authors speculate on the mechanisms linking prenylation to CSC generations through the indirect effect on RAS signaling. However, one the models used in this study (HT29 cell line) is BRAF mutant. What is the possible mechanism of action in this case? Rho signaling can still be responsible in this case, but no results about this are shown. Adding some results on the possible mechanism linking the inhibition of prenylated proteins to the decrease of CSCs would really increase the relevance of the paper.
Response 6: Thank you for your keen comments. As you mentioned, HT29 has BRAF mutation and wild type KRAS, and DLD-1 has KRAS mutation and BRAF wild type. Therefore, inhibition of RAS function in HT29 cells would not be effective. However, in addition to RAS-RAF-ERK signaling, RAS downstream includes mTOR, RalA/B, and Rho signaling, which are associated with cancer stem cells (CSCs) and invasion. Therefore, CSC suppressing effect could be seen by only farnesylation inhibition of RAS in HT29 cells. Moreover, the function of RalA/B and Rho can be also affected by regulation of geranylgeranylation. Therefore, we could see a more prominent CSC suppressing effect by inhibition of geranylgeranylation or both farnesylation/ geranylgeranylation, compared to inhibition of farnesylation alone in Figure 3.
Because Ral A prenylation data also was added in Supplementary Figure, we added “Ral” in some sentences in Discussion section (line 260-262) as follows (red).
“The prenylation of Ras and Ral/Rho proteins via MVA pathway-dependent farnesylation and geranylgeranylation, respectively, is important for their function. The Ras, Ral and Rho families are associated with many tumor characteristics, such as invasive growth, cell survival, and three-dimensional growth3,25”
Point 7. Another relevant point in the discussion is not clear: do the Authors evisage a potential clinical applicability of metformin as a CSC inhibitor? This leads to a more fundamental question: are the concentrations of metformin used in this study clinically relevant and achievable in patients? Moreover, statin effect was not shown to be relevant as single agent in CSCs, but no combination was obtained with metformin. Probably this goes beyond the scope of the paper, but such combination could be a possible approach.
Response 7: Thank you for your practical comment. We used 5-10 mM of metformin in in vitro model, and 250 mg/kg/day of metformin in in vivo model. Those doses are within common ranges used in many references, and using formula for conversion of animal dose to human equivalent dose based on body surface area as follows, we can calculate the human equivalent dose of 1200mg/d for 60kg adult, according to the mouse dose of 250 mg/kg/day of metformin used in our in vivo experiment. The dose of 1200mg/d for adult is within common dose range for diabetes treatment. (image attached word file)
As for the effect of drug combination, as you mentioned, the combination of metformin and other CSC suppressing drugs, including statin might be more effective than single agent, and can decrease dose of each drug and side effect of them.
Point 8. Is there a role for FFTi or GGTi in cancer treatment? Are there any supporting clinical trials ongoing or with results?
Response 8: Two potent farnesyltransferase inhibitors (FTIs), tipifarnib and lonafarnib, showed efficacy in inhibiting tumour growth of preclinical models of HRAS-driven cancers, but they did not show clinical efficacy in KRAS-driven cancers in phase III trials.
As for GGTi, there was a phase I clinical trial for GGTI-2418, but this study was terminated prior to dose expansion based on a sponsor decision [Target Oncol. 2019 Oc;14(5):613-618].
Given feedback signaling of RAS-RAF-ERK pathway and other closely related signaling such as PI3K-AKT-mTOR pathway, it is difficult to find significant role as single agent. Therefore, combination trial should be considered as recent other drug trials.
I think the anti-tumor effect of metformin also should be emphasized in the aspect of combining effect by other direct and indirect anti-tumor mechanisms of metformin, not only through a specific single mechanism like prenylation-dependent or tumor-metabolism related mechanism.
Point 9: Finally, I believe there is a problem with the references, since the first to appear is number 8 (line 45). Moreover, after each reference number follows the author of the referenced article, as if something went wrong with the bibliography manager. Probably, the reference list was set in alphabetical order, but this is not the right format for scientific articles on this journal.
Response 9: Thank you for your important comment. As you mentioned, we changed the reference format as shown in submission instruction of journal.

Reviewer 4 Report
The article “Metformin Suppresses Cancer Stem Cells through AMPK Activation and Inhibition of Protein Prenylation of the Mevalonate Pathway in Colorectal Cancer” by Seo and colab. is interesting and well-written.
I only have few suggestions that might improve it slightly.
- In the Introduction section, I think an additional paragraph detailing the mevalonate pathway and why it is important to cancer cells could be useful for the readers
- Figure 3A is quite difficult to read
- In the IHC Results section (rows 199-202), you describe the results of staining for CSC markers in the metformin group and in the metformin-mevalonate group. How were the IHC scores in the mevalonate-only group when compared with the metformin-only group?
- I think adding a paragraph in which you underline the importance of the study (is it the first of its kind?) and also the study limitations could also improve the article
Minor comments:
- Page 2, row 46 – “Approximately 25% of CRC patients are already stage IV at diagnosis” instead of “Approximately 25% of CRC patients are first diagnosed at stage IV”
- Page 2, row 70 - “and resistance to chemotherapy” instead of „and chemotherapeutic resistance”
Author Response
Response to Reviewer 4 Comments
Referee # 4 :
The article “Metformin Suppresses Cancer Stem Cells through AMPK Activation and Inhibition of Protein Prenylation of the Mevalonate Pathway in Colorectal Cancer” by Seo and colab. is interesting and well-written.
I only have few suggestions that might improve it slightly.
Point 1. In the Introduction section, I think an additional paragraph detailing the mevalonate pathway and why it is important to cancer cells could be useful for the readers
Response 1: Thank you for your kind comment. We added some sentences in Introduction section (line 69-71) for explaining the mevalonate pathway in detail as follows.
“The Mevalonate (MVA) pathway leads to the synthesis of sterols and protein prenylation that has important roles in tumor-growth. Multiple enzymes of this pathway are essential for proliferation and survival of various types of cancer cells, and are up-regulated in many cancers.”
Point 2. Figure 3A is quite difficult to read
Response 2: Thank you for your comment. We changed it into a larger figure.
Point 3. In the IHC Results section (rows 199-202), you describe the results of staining for CSC markers in the metformin group and in the metformin-mevalonate group. How were the IHC scores in the mevalonate-only group when compared with the metformin-only group?
Response 3: Thank you for your kind comment. We found significantly higher IHC scores of mevalonate-only group, compared to the metformin-only group, and marked it in Figure 4D.
Point 4. I think adding a paragraph in which you underline the importance of the study (is it the first of its kind?) and also the study limitations could also improve the article
Response 4: Thank you for your kind comment. As your comment, our finding is the first report on this field, and it was mentioned in the last paragraph in Introduction section.
And, we added sentences about limitation in the last paragraph of Discussion section as follows (line 292-296).
“However, because metformin has many molecular mechanisms of anti-tumor effect, we could not elucidate the detailed interaction between prenylation-dependent and other direct and indirect mechanisms of metformin-induced anti-tumor or CSC suppression. Further studies would be needed to investigate these cross-reactions among direct and indirect mechanisms of CSC suppressing effect of metformin.”
Minor comments:
- Page 2, row 46 – “Approximately 25% of CRC patients are already stage IV at diagnosis” instead of “Approximately 25% of CRC patients are first diagnosed at stage IV”
- Page 2, row 70 - “and resistance to chemotherapy” instead of „and chemotherapeutic resistance”
Thank you very much for your correction. We corrected those sentences as your comment (line 46, line 66).

Round 2
Reviewer 3 Report
The authors replied to all major concerns and integrated the manuscript with the suggested comments by the reviewers.